# PAI-1 Regulation of p53 Expression and Senescence in Type II Alveolar Epithelial Cells

**DOI:** 10.3390/cells12152008

**Published:** 2023-08-05

**Authors:** Tapasi Rana, Chunsun Jiang, Sami Banerjee, Nengjun Yi, Jaroslaw W. Zmijewski, Gang Liu, Rui-Ming Liu

**Affiliations:** 1Division of Pulmonary, Allergy, and Critical Care, Department of Medicine, University of Alabama at Birmingham, Birmingham, AL 35294, USA; 2Department of Biostatistics, University of Alabama at Birmingham, Birmingham, AL 35294, USA

**Keywords:** PAI-1, proteasome activity, p53 degradation, ATII cell senescence

## Abstract

Cellular senescence contributes importantly to aging and aging-related diseases, including idiopathic pulmonary fibrosis (IPF). Alveolar epithelial type II (ATII) cells are progenitors of alveolar epithelium, and ATII cell senescence is evident in IPF. Previous studies from this lab have shown that increased expression of plasminogen activator inhibitor 1 (PAI-1), a serine protease inhibitor, promotes ATII cell senescence through inducing p53, a master cell cycle repressor, and activating p53-p21-pRb cell cycle repression pathway. In this study, we further show that PAI-1 binds to proteasome components and inhibits proteasome activity and p53 degradation in human lung epithelial A549 cells and primary mouse ATII cells. This is associated with a senescence phenotype of these cells, manifested as increased p53 and p21 expression, decreased phosphorylated retinoblastoma protein (pRb), and increased senescence-associated beta-galactose (SA-β-gal) activity. Moreover, we find that, although overexpression of wild-type PAI-1 (wtPAI-1) or a secretion-deficient, mature form of PAI-1 (sdPAI-1) alone induces ATII cell senescence (increases SA-β-gal activity), only wtPAI-1 induces p53, suggesting that the premature form of PAI-1 is required for the interaction with the proteasome. In summary, our data indicate that PAI-1 can bind to proteasome components and thus inhibit proteasome activity and p53 degradation in ATII cells. As p53 is a master cell cycle repressor and PAI-1 expression is increased in many senescent cells, the results from this study will have a significant impact not only on ATII cell senescence/lung fibrosis but also on the senescence of other types of cells in different diseases.

## 1. Introduction

Cellular senescence, which occurs in both proliferative cells and post-mitotic cells, has been increasingly recognized as an important mechanism for aging and aging-related diseases [1,2,3,4,5]. Cellular senescence, especially the senescence of stem cells, contributes importantly to the pathophysiology associated with aging and aging-related diseases, including idiopathic pulmonary fibrosis (IPF), a progressive fetal lung disorder. Alveolar type II (ATII) cells can self-renew and differentiate into type I alveolar epithelial cells and therefore are considered stem cells or alveolar progenitors [6,7]. ATII cell senescence is evident in fibrotic lung diseases, including IPF [8,9,10,11,12], and in experimental fibrosis models [13,14,15,16]. The molecular mechanism underlying ATII cell senescence in IPF lungs, however, is not fully understood.

p53 is a master regulator of the cell cycle and a universal inducer of cell senescence. Increased expression of p53 is observed in senescent cells in many types of fibrotic diseases [17,18,19]. Indeed, the p53 pathway is among the most dysregulated pathways identified in IPF lungs [15,20,21]. Although it is well documented that p53 undergoes a variety of posttranslational modifications that lead to the stabilization/destabilization of this protein in cancer cells, how p53 is activated in ATII cells in the fibrotic lung [22,23] is unclear. In previous studies, we showed that the increased expression of plasminogen activator inhibitor 1 (PAI-1), a serine protease inhibitor playing a critical role in the development of lung fibrosis [24,25,26,27], contributes importantly to ATII cell senescence in the fibrotic lung [15,28]. We also showed that PAI-1 expression increases with age in mouse ATII cells [29] and that PAI-1 positively regulates p53 expression and promotes ATII cell senescence through activating the p53-p21-pRb cell cycle repression pathway, although the mechanism underlying PAI-1 induction of p53 remains to be determined [15]. As p53 is a master cell cycle repressor and PAI-1 expression is increased in many senescent cells [15,28,30,31,32,33,34], uncovering the molecular mechanism by which PAI-1 regulates p53 expression will have a significant impact on multiple diseases.

In this study, we show for the first time that PAI-1 binds to proteasome components and inhibits the activity of proteasomes in human lung epithelial A549 cells and primary mouse ATII cells. This is associated with an inhibition of p53 degradation and cell senescence. We also show that, although overexpression of a mature form of PAI-1 protein intracellularly alone induces ATII cell senescence, just like overexpression of wild-type PAI-1 protein, it does so through a p53-independent pathway. As PAI-1 expression is increased in IPF lungs and experimental lung fibrosis models, our results suggest that increased PAI-1 may underlie the dysregulation of p53 and ATII cell senescence in IPF lungs.

## 2. Materials and Methods

### 2.1. Mouse ATII Cell Isolation

Mouse lung AT2 cells were isolated from adult wild-type C57/B6 and PAI-1 deficient (PAI-1^−/−^) mice (6–12 weeks old) according to the protocol we described earlier [15,28]. Briefly, mouse lungs were digested with solution containing 300 U/mL collagenase type I (Worthington Lakewood, NJ, USA, LS004196), 4 U/mL elastase (Worthington, LS002292), 5 U/mL dispase (Worthington, LS02104), and 100 mg/mL DNase I in a Hanks’ balanced salt solution in a 37 °C water bath for 25 min and then incubated at 37 °C for another 20 min. Then, tissues were dissociated by pipetting up and down, and single-cell suspensions were prepared by passing the lung digestion solutions through a 40 mm mesh cell strainer. Red blood cells were lysed using a red lysis buffer, and then macrophages and lymphocytes were removed via incubation with biotinylated rat anti-mouse CD45 and CD16/32 (BD Biosciences San Jose, California, USA), respectively. Cells were then cultured with Dulbecco’s modified Eagle medium/Nutrient Mixture F-12 (DMEM/F-12, 10-013-CV, Corning/21127-022; GibCo Montana, CA, USA) medium containing 10% fetal bovine serum (FBS) in a 37 °C cell culture incubator overnight to remove fibroblasts. Suspended cells were harvested and used for the experiments. The purity of ATII cells isolated using this protocol is >90%, as we demonstrated previously [15]. All of the experiments were conducted with primary ATII cells within 5–6 days after isolation. All procedures involving animals were approved by the institutional animal care and use committees at the University of Alabama at Birmingham (UAB) and conducted at the UAB animal facilities under specific pathogen–free conditions.

### 2.2. Construction of Wild-Type and Mutant PAI-1 Lentiviruses

Mouse fibroblast cDNA was made using an iScript cDNA synthesis kit (BIO-RAD Hercules, CA, USA) from respective total RNA. PAI-1 full-length cDNA was PCR amplified using the following primer set forward primer 5′-ATGCAGATGTCTTCAGCCCTTGCTT-3′ and reverse primer 5′-TCAAGGCTCCATCACTTGGCCCA-3′. PCR amplified product was eluted from agarose gel and purified. Then, another set of primers—5′-GGAATTCGCCACCATGCAGATGTCTTCAGCCCTTGCTT-3′ and 5′-GCGGATCCTCAAGGCTCCATCACTTGGCCCA-3′ containing EcoRI and BamH1 restriction enzyme site and KOZAC sequence—were used to clone into lentivirus vector pCDH-CMV-MCS-EF1-copGFP (System Biosciences Cat# CD511B-1 Palo Alto, CA, USA). The full-length sequence was confirmed through sequence analysis. Truncated mouse PAI-1 cDNA was PCR amplified from the above PAI-1 lentivirus vector using forward primer 5’-CTCGGATCCGCCACCATG TTCACTTTACCCCTCCGAG-3’ and reverse primer 5’-CGGCGGCCGCTCAAGGCTCCATCACTTGGCCC-3’. The PCR product was digested by BamHI and NotI and cloned into an empty pCDH-CMV-MCS-EF1-copGFP lentivirus vector (Cat #CD511B-1; System Biosciences Palo Alto, CA, USA). The correct truncated mouse PAI-1 sequence was confirmed by DNA sequencing.

### 2.3. Cell Culture and Treatments

A549 cells, isolated from the lung of a cancer patient, display lung epithelial phenotype. A549 cells were cultured with Ham’s F-12 medium (21127-022, GibCo, CA, USA) containing 10% FBS and antibiotics. The cells were transfected with non-target siRNA or PAI-1 siRNA (SC36179; Santa Cruz, Dallas, TX, USA) for 24 h and then treated with bleomycin (5 mU/mL, 203401; Calbiochem, San Diego, CA, USA) for another 24 h, followed by culturing in a serum-free and bleomycin-free medium for 24 hrs. ATII cells were treated with 5 mU/mL of bleomycin in a serum-free medium for 24 h and then cultured in a bleomycin-free and serum-free medium for an additional 48 h. For transduction with the PAI-1 expressing lentivirus or control virus, A549 cells and AT2 cells were cultured in complete cell culture media. Virus-containing medium was removed after 24 h of transduction, and the cells were cultured in a virus-free medium for another 48 h.

### 2.4. Western Blot Analysis

Cells were lysed with cell lysis buffer containing 25 mM Tris•HCl pH 7.6, 150 mM NaCl, 1% NP-40, 1% sodium deoxycholate, 0.1% SDS, and protease inhibitor (P8340; Sigma Aldrich, Saint Louis, MO, USA) and phosphatase inhibitor cocktails (P5726; Sigma). Cell lysates with 50 μg of proteins were resolved using 10% SDS-PAGE gel and blotted onto polyvinylidene fluoride (PVDF) membranes. The membranes were probed with the following antibodies: PAI-1 (ASMPAI-GF; Molecular Innovation Novi, MI, USA), p53 (SC-6243; Santa Cruz), p21 (SC-397; Santa Cruz), phosphorylated retinoblastoma protein (pRB) (CN 8180; Cell Signaling Technology Danvers, MA, USA), mouse monoclonal anti-20S proteasome alpha 3 subunit (SC-166205; Santa Cruz), polyclonal rabbit anti-19S antibody (BML-PW8250-0025; Enzo Farmingdale, NY, USA), and anti-GAPDH (SC47724; Santa Cruz). The secondary antibodies used for Western were goat anti-rabbit IgG–Peroxidase tagged (A9169; Sigma-Aldrich Saint Louis, MO, USA) and anti-mouse IgG secondary antibody (catalog# 223-005-024; Jackson ImmunoResearch West Grove, PA, USA). The protein bands were visualized using an enhanced chemiluminescence detection reagent (WBKLS0500 Millipore Sigma Saint-Louis, MO, USA; Chemiluminescent ECL Substrate), semi-quantified using Image J software (Image J 1.53c, https://imagej.nih.gov/ij/, last accessed on 29 October 2020), and normalized by the corresponding GAPDH band.

### 2.5. Determination of the Activity of Senescence-Associated Beta Galactosidase

The activity of senescence-associated beta galactosidase (SA-β-gal) was determined using 5-bromo-4-chloro-3-indolylb-D-galactoside (X-gal, CN, 4063-102; MP Biomedicals Santa Ana, CA, USA), following the protocol we described previously [15,28]. Briefly, cells were fixed in 2% formaldehyde plus 0.2%glutaraldehyde in PBS at room temperature for 10 min, rinsed with PBS, and then incubated overnight in SA-β-gal staining solution containing 1 mg/mL X-gal. The images were taken with 10× magnification using a Zeiss AxioCamMR microscope (Germany). Total cells and SA-β-gal positive cells were counted for each image (10 images/per sample and 4–12 samples/group). The results are expressed as the percentages of total cells.

### 2.6. ELISAs of Senescence Markers in the Conditional Medium (CM)

The cells (A549 and ATII cells) were treated with bleomycin for 24 h and then cultured in a serum-free and bleomycin-free medium for another 24 h (A549 cells) or 48 h (ATII cells). The conditional media (CM), which do not contain bleomycin or serum, were then collected, and the aliquots were stored at −80 °C. ELISAs were performed to determine senescence markers, Interleukin 6 (IL-6) and insulin-like growth factor binding protein 3 (IGFBP3) in the CM from AT2/A549 cells using ELISA kits from R&D Systems MN, USA, according to the manufacturer’s protocols.

### 2.7. Immunoprecipitation

Cell lysates containing 200 μg of protein were precleared using preimmune IgG plus protein A/G agarose beads, and the supernatants were immunoprecipitated with monoclonal antibody to PAI-1 (MA5-17171; Invitrogen) Waltham, MA, USA, proteasome 20S alpha subunit (SC166205; Santa Cruz Dallas, TX USA), or proteasome 19S subunit (BML-PW8765-0025; Biomol/ENZO PA, USA) and a 50% slurry of protein A agarose beads overnight at 4 °C. After washing with the buffer containing 50 mmol/L Tris, pH 7.5; 150 mmol/L NaCl; 1% NP-40; and 0.5% deoxycholate and protease inhibitors, proteins were released and separated using 10% Bis-Tris gels and proteins transferred to PVDF membranes. The membranes were blotted with antibodies to proteasome 20S alpha subunit, 19S subunit, or PAI-1, followed by the corresponding secondary antibody. Proteins on the membrane were stained with Ponceau S to show equal protein loading after pulldown (GAPDH or β-actin was not pulled down).

### 2.8. Proteasome Activity Assay

Cells were lysed using lysis buffer containing 25 mmol/L Tris-HCl, pH 7.5; 100 mmol/L KCl; 1 mmol/L EDTA; 0.1% (*v*/*v*) Triton X-100; and 1 mmol/L phenylmethane sulfonylfluoride. Cell lysates (10 μg protein/each) were resuspended in 25 mmol/L Tris-HCl, pH 7.4; 1 mmol/L dithiothreitol; and 20 mmol/L KCl. The proteasome activities were measured with 100 μmol/L each of fluorogenic substrates: Z-Leu-Leu-Glu-7-amino-4- methylcoumarin (AMC) for caspase-like, Suc-Leu-Leu-Val-Tyr-AMC for chymotrypsin-like, and Boc-Leu-Ser-Thr-Arg-AMC (Sigma-Aldrich Saint Louis, MO, USA) for trypsin-like activity in the absence or presence of 25 μmol/L proteasome inhibitor MG-132 (Selleckchem, S2619, Houston, TX, USA). The change in fluorescence was monitored for 2 h at 37 °C in a microplate fluorescence reader with an excitation wavelength of 380 nm and emission wavelength of 460 nm. The activity was calculated after subtracting the background (part not inhibited by MG-132).

### 2.9. Analysis of p53 Protein Degradation Rate

Cycloheximide chase techniques were used to determine the degradation rates of the p53 protein. Briefly, A549 cells were treated with bleomycin (5 mU/mL) with or without PAI-1 inhibitor TM5275 (25 μM) or treated with bleomycin after transduced with lentiviruses expressing wild-type PAI-1 or virus vehicle for 24 h. Bleomycin, TM5275, and viruses were removed, and the cells were cultured in a fresh medium for another 24 h and then treated with 50 μg/mL cycloheximide (CHX) (catalog #01810; Sigma-Aldrich, Saint Louis, MO, USA) for 0, 2, 4, or 8 h. The cells were harvested, and Westerns were performed to determine the degradation rates of p53 and p21 (control) proteins. For studying the p53 degradation rate in ATII cells, ATII cells isolated from WT and PAI-1^−/−^ mice were treated with bleomycin for 24 h and then cultured in a bleomycin-free medium for another 48 h, and then cells were treated with CHX (10 μg/mL) for 0, 30, 60, and 120 min in the media containing 1% FBS.

### 2.10. Statistics

All of the data are expressed as means ±SD (*n* ≥ 3) unless otherwise indicated. Student’s *t*-test was used for two-group comparison, whereas one-way ANOVA was used for multiple-group comparison, and the post-hoc analysis was performed using Tukey’s test. The p53 and p21 protein degradation rates between different treatment groups were compared using generalized additive models. The R package mgcv was used to set up and fit the generalized additive models and to summarize the results. The marginal effects of the predictor ‘time’ were calculated to show how the outcome changes over time.

## 3. Results

### 3.1. PAI-1 Binds to Proteasome Components in A549 Cells

To determine whether PAI-1 promotes p53 expression by modulating the proteasome activity and thus p53 degradation, we first examined if PAI-1 can directly bind to proteasome components. Human lung epithelial A549 cells were transfected with PAI-1 siRNA or non-targeted (NT) siRNA or transduced with lentivirus that expresses wild-type PAI-1 protein or virus vector and then treated with bleomycin. The binding of PAI-1 protein to proteasome components 20S alpha 3 (20S α3) and 19S Rpt3/S6b subunits was studied via immunoprecipitation and Westerns. The results show that treatment with bleomycin increased the binding of PAI-1 protein to 20Sα3 and 19S Rpt3/S6b subunits (Figure 1A–C). Silencing PAI-1, on the other hand, significantly reduced bleomycin-induced binding (Figure 1A–C). Overexpression of wild-type PAI-1 protein alone increased the binding of PAI-1 to 20S α3 and 19S Rpt3/S6b, although it did not further increase bleomycin-stimulated binding (Figure 1D–F). Our data demonstrate for the first time that PAI-1 protein binds to proteasome components in A549 cells. The data suggest that PAI-1 may directly modulate proteasome functions.

### 3.2. PAI-1 Inhibits the Proteasome Caspase-Like Activity in A549 Cells

To further assess the interactions between PAI-1 protein and proteasome components, the caspase-like, chymotrypsin-like, and trypsin-like activities of the proteasome were measured using specific substrates and confirmed with proteasome inhibitor MG132. The results show that treatment with bleomycin suppressed the caspase-like and chymotrypsin-like activities, although it had no significant effect on the trypsin-like activity of the proteasome in A549 cells (Figure 2A–F). Silencing PAI-1, on the other hand, restored the caspase-like activity that was inhibited by bleomycin (Figure 2A,B), although it had no significant effect on bleomycin-mediated inhibition of the chymotrypsin-like activity (Figure 2C,D). Overexpression of PAI-1 protein alone suppressed the caspase-like and chymotrypsin-like activities (Figure 2G–J) but had no significant effect on the trypsin-like activity (Figure 2K,L). Treatment with bleomycin, however, did not further inhibit the caspase-like or chymotrypsin-like activity on top of PAI-1 overexpression (Figure 2G–J). These results support the notion that PAI-1 interacts with proteasome components and inhibits its function.

### 3.3. PAI-1 Suppresses p53 Protein Degradation in A549 Cells

To determine whether p53 protein degradation is inhibited because of increased PAI-1 expression and the suppression of proteasome activity, p53 protein stability was assayed via cycloheximide chase experiments. The results show that bleomycin treatment significantly decreased the p53 protein degradation rate, although it had no significant effect on the p21 protein degradation rate (the time-dependent changes in the amounts of p53 and p21 proteins between NTsiRNA control and NTsiRNA bleomycin group were statistically significant for p53, *p* = 0.000103, but not for p21, *p* = 0.258) in A549 cells (Figure 3A–C). Silencing PAI-1 alone had no significant effect on the p53 degradation rate (time-dependent changes between NTsiRNA control and PAI-1siRNA control were not statistically significant, *p* = 0.219) but significantly attenuated bleomycin-induced suppression of the p53 degradation rate (the time-dependent changes in the amounts of p53 protein between NTsiRNA control and PAI-1siRNA control were not statistically significant, *p* = 0.219, but were significant between PAI-1siRNA bleomycin and NTsiRNA bleomycin groups, *p* = 0.0118) (Figure 3A,B). Silencing PAI-1 had no significant effect on p21 protein degradation (Figure 3A,C). We also found that the inhibition of PAI-1 activity with a small molecule PAI-1 inhibitor TM5275 almost completely restored the p53 protein degradation rate that was inhibited by bleomycin in A549 cells (the time-dependent changes in the amounts of p53 protein between bleomycin + TM5275 and bleomycin + solvent were highly significant, *p* = 0.00046; those between bleomycin + TM5275 and saline + TM5275 were not significant, *p* = 0.626; those between bleomycin +TM5275 and saline + solvent were not significant, *p* = 0.945) (Figure 3D,E). Moreover, we found that overexpression of PAI-1 protein alone significantly suppressed the p53 protein degradation rate (the time-dependent changes in the amounts of p53 protein between PAI-1 virus and control virus groups were statistically significant, *p* = 0.05) and potentiated bleomycin-mediated suppression of p53 degradation rate (the time-dependent changes in the amounts of p53 protein between PAI-1 virus + bleomycin and PAI-1 virus only were statistically significant, *p* = 0.00829), although overexpression of PAI-1 had no significant effects on p21 protein degradation rate (Figure 3F–H).

### 3.4. Increased PAI-1 Mediates Bleomycin-Induced p53 Expression and Senescence in A549 Cells

Associated with the suppression of proteasome activity and p53 degradation, treatment with bleomycin was shown to increase the amount of PAI-1 protein as well as p53 and p21 proteins in A549 cells (Figure 4A–D). This is associated with increases in the activity of senescence-associated beta-galactose (SA-β-gal) (Figure 4E,F) as well as the secretion of IL-6 and IGFBP3 (Figure 4F,G), two senescence markers. Silencing PAI-1, on the other hand, significantly reduced bleomycin-induced p53 and p21 expression, as well as the activity of SA-β-gal and the secretion of IL-6 and IGFBP3 (Figure 4). These data suggest that increased PAI-1 mediates at least in part bleomycin-induced p53 expression and senescence in A549 cells. Moreover, we showed that overexpression of wild-type PAI-1 (wtPAI-1) alone (Figure 5A,B) increased p53 and p21, suppressed retinoblastoma protein phosphorylation (pRb) (Figure 5A,C–E), and increased SA-β-gal activity as well as the secretion of IL-6 and IGFBP3 (Figure 5F–I), indicating that overexpression of PAI-1 alone induces A549 cell senescence. Overexpression of wtPAI-1 also significantly enhanced bleomycin-induced p53 expression (Figure 5A,C) as well as bleomycin-induced senescence response in A549 cells (Figure 5). Together, the results suggest that increased PAI-1 not only mediates at least in part bleomycin-induced p53 and cell senescence but can also promote A549 cell senescence directly.

### 3.5. PAI-1 Binds to Proteasome Components, Inhibits Proteasome Activity and p53 Degradation, and Induces Senescence in Primary Mouse ATII Cells

To further determine whether PAI-1 also induces cell senescence through suppressing proteasome function and thus p53 degradation in primary cells, mouse ATII cells were isolated from wild-type (WT) and PAI-1 deficient (PAI-1^−/−^) mice and treated with bleomycin in vitro. The results show that treatment with bleomycin increased the binding of PAI-1 to 20S α3 subunit (Figure 6A,B) and inhibited the activities of all three proteasome enzymes in ATII cells (Figure 6C–H). Deletion of PAI-1, on the other hand, eliminated bleomycin-induced binding of PAI-1 to 20S α3 subunit and restored in part the caspase-like and tyrosine-like proteasome activities, although it had no significant effect on bleomycin-induced suppression of chymotrypsin-liked activity (Figure 6A–H). Deletion of PAI-1 also reversed bleomycin-mediated suppression of p53 degradation (the time-dependent changes in the amounts of p53 protein were statistically significant between PAI-1^+/+^ and PAI-1^−/−^ ATII cells), although deletion of PAI-1 had no significant effect on p21 degradation, in ATII cells (Figure 6I–K). This was associated with a significant reduction of bleomycin-induced expression of p53 and p21 (Figure 7A–D), the SA-β-gal activity (Figure 7E,F), and IL-6 secretion (Figure 7G). Together, the results suggest that PAI-1 can bind to the proteasome and inhibit its activity and p53 degradation, leading to ATII cell senescence.

### 3.6. Intracellular Mature PAI-1 Induces Senescence but Has No Significant Effect on p53 Expression in Primary Mouse ATII Cells

PAI-1 is a secreted protein that is believed to function extracellularly, although emerging evidence suggests that intracellular PAI-1 also has some important functions [35,36,37]. To test whether intracellular PAI-1 is involved in ATII cell senescence, primary ATII cells were isolated from PAI-1^−/−^ mice and transduced with lentivirus expressing wild-type PAI-1 (wtPAI-1), secretion-deficient PAI-1 (sdPAI-1, secretion signal removed), or virus vehicle only. The results show that the infection of PAI-1^−/−^ ATII cells with lentivirus expressing wtPAI-1, but not sdPAI-1, significantly increased PAI-1 protein level in the conditional medium (extracellular) (Figure 8A), which led to an inhibition of plasmin activity in the conditional medium (Figure 8B). Our data also show that both wtPAI-1 and sdPAI-1 viruses increased the intracellular PAI-1 protein significantly, although the wtPAI-1 virus mainly expressed a premature form of PAI-1 (MW 45 kd), while the sdPAI-1 virus expressed a mature form of PAI-1 (MW 42 kd) (Figure 8C,D). The data confirm that we successfully constructed wtPAI-1 and sdPAI-1 lentiviruses. Importantly, we find that overexpression of wtPAI-1 increased p53 and p21 expression as well as SA-β-gal activity, whereas overexpression of sdPAI-1 had no significant effect on the expression of either p53 or p21, although it increased SA-β-gal activity (Figure 8C–H). These results suggest that the intracellular mature form of PAI-1 can induce ATII cell senescence, but it does so through a p53-p21 independent pathway.

## 4. Discussion

Emerging evidence indicates that cellular senescence contributes importantly to the pathophysiology of aging and aging-related diseases [1,2,3,4,5]. The mechanisms underlying cellular senescence during aging and in aging-related diseases, however, remain poorly understood. PAI-1 is a serine protease inhibitor playing a major role in hemostasis. Besides suppression of fibrinolysis, PAI-1 has many other functions, including the modulation of cell adhesion, migration, and proliferation, dependent or independent of its protease inhibitory activity [38]. Notably, PAI-1 expression is increased with age and in many aging-related diseases [39,40,41,42,43,44]. Emerging evidence, including that from this lab, shows that PAI-1 plays a critical role in cell senescence, although the mechanism by which PAI-1 promotes cell senescence remains obscure. In this study, we show for the first time that PAI-1 binds to proteasome components and inhibits the proteasome activity and the degradation of p53 protein, a master cell cycle repressor, in human lung epithelial A549 cells and \ primary mouse ATII cells. This is associated with the increase in p53 expression and cell senescence phenotype. Our data also show that, like extracellular PAI-1, intracellular mature PAI-1 protein can also induce ATII cell senescence but through a p53-independent mechanism. The results from this study reveal a novel mechanism by which PAI-1 modulates the expression of the master cell cycle repressor p53 and promotes cell senescence in alveolar epithelial cells.

Increased PAI-1 has been used as a marker of cell senescence in many in vitro and in vivo studies. The results from our studies and from others further show that increased PAI-1 is not merely a marker but also a mediator of cell senescence in various types of cells [15,28,30,31,33,34,44]. Although different mechanisms have been proposed, studies, including ours, suggest that PAI-1 promotes cell senescence at least in part through inducing p53 [15,33,44], a master controller of the cell cycle. Ghosh et al. showed that TM5441, a potent small molecule inhibitor of PAI-1, effectively prevented doxorubicin-induced senescence in cardiomyocytes, fibroblasts, and endothelial cells, associated with a suppression of doxorubicin-induced expression of p53, p21, and p16 [33]. In a previous study, we showed that specific ablation of PAI-1 in alveolar type II (ATII) cells in mice attenuated bleomycin-induced p53 expression and ATII cell senescence in vivo [15]. Using pharmacological and genetic approaches, we further showed that silencing PAI-1 or inhibiting PAI-1 activity significantly reduced bleomycin- and doxorubicin-induced p53 expression and senescence in rat lung epithelial (L2) cells, whereas silencing p53 dramatically attenuated PAI-1 protein-induced L2 cell senescence [15]. Together, the data suggest that PAI-1 promotes ATII cell senescence at least in part through inducing p53 and activating p53-p21-pRb cell cycle repression pathways [15]. Cell senescence, including senescence of astrocytes, the most abundant type of cells in the brain, is evident in Alzheimer’s disease patients. In a recent study, we showed that H_2_O_2_ induced PAI-1 and p53 as well as SA-β-gal activity in primary human and mouse astrocytes [44]. Inhibition, silencing, or deletion of PAI-1 significantly attenuated H_2_O_2_-induced p53 expression and senescence in both human and mouse astrocytes [44], suggesting that an increase in PAI-1 mediates H_2_O_2_-induced astrocyte senescence probably through inducing p53, although the mechanism underlying PAI-1 induction of p53 remained unclear at the moment.

Tumor suppressor p53 is a master controller of the cell cycle and inducer of cell senescence. Accumulation of p53 is observed in senescent cells in many types of fibrotic pathologies [17,18,19]. Although it is well known that p53 undergoes a variety of posttranslational modifications that leads to stabilization/destabilization of this protein in cancer cells, it is unclear how p53 is activated in ATII cells in fibrotic lung [22,23]. p53 protein is degraded in proteasome [45,46,47]. PAI-1 has been shown to bind to the intracellular α3 subunit of proteasome and suppress the degradation of p53 and IκBα in endothelial cells [48]. In this study, we show, for the first time, that PAI-1 binds to the 20S α3 subunit and 19S Rpt3/S6b subunit of the proteasome in A549 cells and ATII cells. This is associated with an inhibition of caspase-like activity in A549 cells but caspase-like and tyrosine-like activities in ATII cells. Moreover, we show that silencing PAI-1 or deletion of PAI-1 attenuated bleomycin-induced suppression of proteasome activity and promoted p53 degradation, whereas overexpression of PAI-1 suppresses proteasome activity and p53 protein degradation in A549 and ATII cells. Together, our data suggest that PAI-1 increases p53 expression and ATII cell senescence at least in part through binding to proteasome components and inhibiting its activity and thereby p53 degradation.

PAI-1 is a secreted protein that is believed to function mainly in the extracellular space, although emerging evidence suggests that intracellular PAI-1 also has important functions [35,37,48]. To define the role of intracellular PAI-1 in ATII cell senescence, we constructed lentiviruses that express secretion-deficient PAI-1 (sdPAI-1) or wild-type PAI-1 (wtPAI-1). We found that the transduction of PAI-1^−/−^ ATII cells with lentiviruses expressing wtPAI-1 or sdPAI-1 alone induced ATII cell senescence (increased SA-β-gal activity). However, sdPAI-1 did not affect p53 or p21 expression, although wtPAI-1 significantly increased the expression of both p53 and p21. These data suggest that, although intracellular PAI-1 can induce ATII cell senescence, it does so through a p53-independent pathway. It should be pointed out, however, that PAI-1 protein expressed through wtPAI-1 lentiviruses is a premature form (molecular weight 45 kd), while that expressed by sdPAI-1 lentivirus is a matured form as the secretion signal peptide has been genetically removed (molecular weight 42 kd, Figure 8C [44]). Therefore, it is speculated that the premature form of PAI-1 is required for its binding to the proteasome components. More studies are needed to address this issue. Nonetheless, the data presented in the current study further support the notion that intracellular PAI-1 has important functions and can induce cell senescence. More studies are warranted to elucidate the molecular mechanism by which intracellular PAI-1 exerts its biological functions.

In summary, we show for the first time that PAI-1 binds to proteasome components and inhibits the proteasome activity and the degradation of the master cell cycle repressor p53 in ATII cells, which is associated with the increased p53 expression and ATII cell senescence. We also show that intracellular mature PAI-1, like extracellular PAI-1, can induce ATII cell senescence. However, this mature PAI-1 protein induces cell senescence through a p53-independent mechanism. The results from this study reveal a novel mechanism by which PAI-1 modulates the expression of the master cell cycle repressor p53 and promotes ATII cell senescence.

## Figures and Tables

**Figure 1 cells-12-02008-f001:**
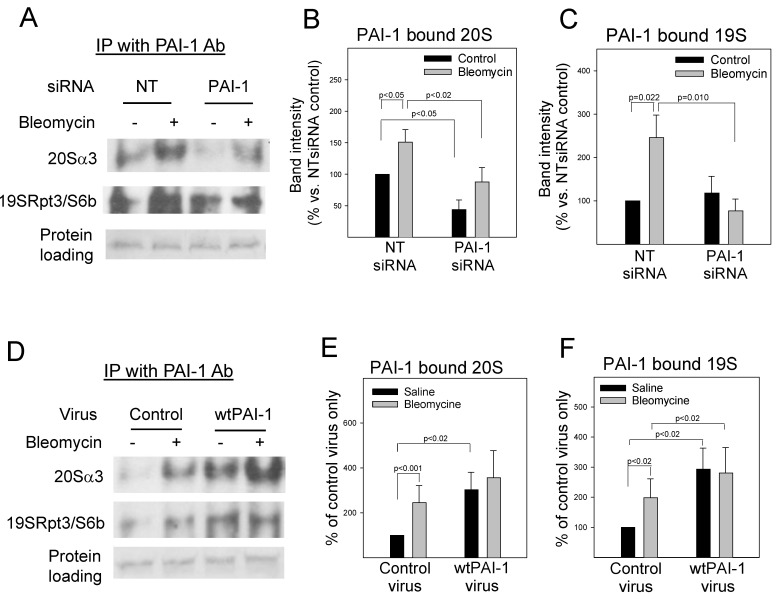
PAI-1 binds to proteasome components in A549 cells. (**A**,**D**) Immunoprecipitation-immunoblotting analysis of PAI-1 interaction with proteasome 20S α3 and 19S Rpt3/S6b subunits in A549 cells. A549 cells were transfected with PAI-1 siRNA/non-target (NT) siRNA or transduced with control/PAI-1 expressing viruses and then treated with bleomycin. PAI-1 protein was immunoprecipitated with anti-mouse PAI-1 monoclonal antibody, and Westerns were conducted with specific antibody to 20Sα3 or 19SRpt3/S6b. Proteins on the membrane were stained with Ponceau S to show equal sample loading. (**B**,**C**,**E**,**F**) Semi-quantification of the band intensities, normalized by the corresponding protein staining band *(n* = 3).

**Figure 2 cells-12-02008-f002:**
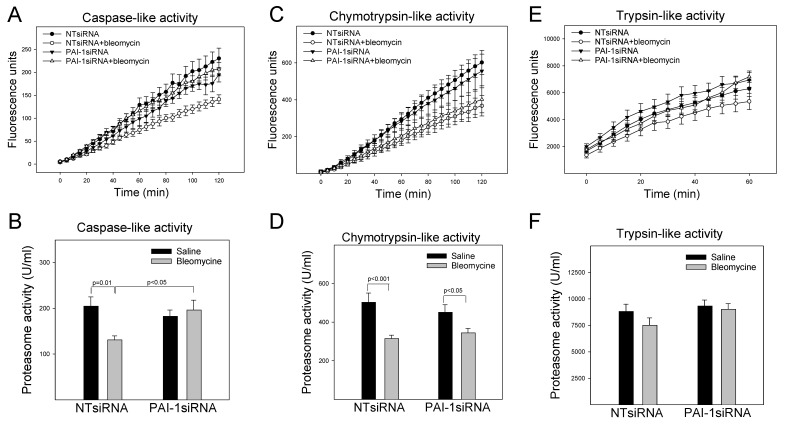
PAI-1 inhibits proteasome activity in A549 cells. (**A**–**F**) Effects of silencing PAI-1 on the activities of the proteasome enzymes. (**G**–**L**) Effects of overexpression of PAI-1 on the activities of proteasome enzymes. A549 cells were transfected with PAI-1siRNA or transduced with PAI-1 expressing viruses and then treated with bleomycin, as described in Figure 1. The activities of three proteasome enzymes were measured using special substrates, as described in the method section. **Top panels:** Representative graphic pictures of fluorescence changes with time. **Bottom panels:** Quantitative data of the proteasome enzymes (n = 4–6).

**Figure 3 cells-12-02008-f003:**
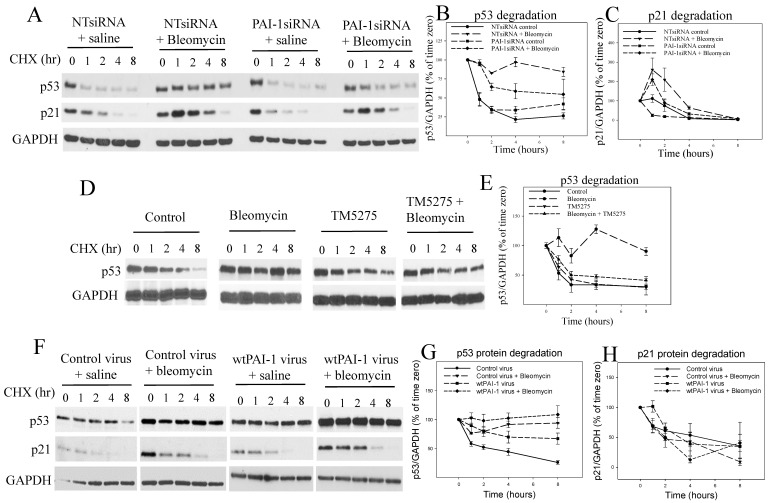
PAI-1 suppresses p53 degradation in A549 cells. (**A**–**C**) Effects of silencing PAI-1 on p53 degradation in A549 cells. (**D**,**E**) Effects of a small molecule PAI-1 inhibitor TM5275 on p53 degradation in A549 cells. (**F**–**H**) Effects of PAI-1 overexpression on p53 degradation in A549 cells. A549 cells were transfected with PAI-1siRNA, pretreated with TM5275, or transduced with PAI-1 expressing viruses and then treated with bleomycin for 24 h before adding cycloheximide (CHX) as described in the Material and Methods section. Cells were harvested at different times after CHX treatment. The amounts of p53, p21 (for comparison), and GAPDH (loading control) proteins were assessed using Westerns. The results were normalized by GAPDH and expressed as percentages of time zero (n = 3–4).

**Figure 4 cells-12-02008-f004:**
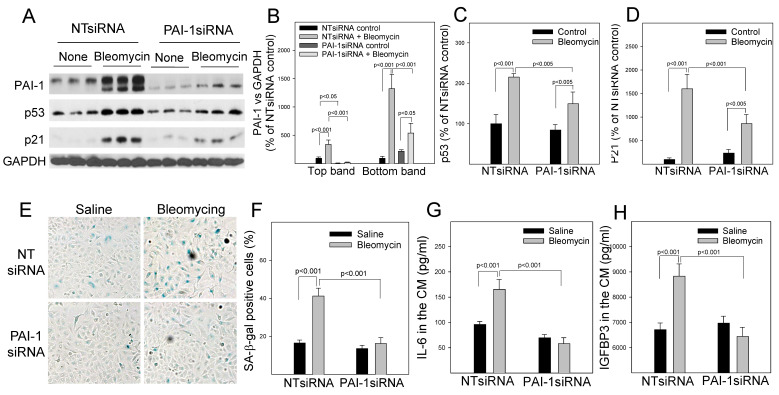
Silencing PAI-1 attenuates bleomycin-induced senescence in A549 cells. (**A**–**D**) Western analyses and quantification of PAI-1, p53, and p21. A549 cells were transfected with PAI-1 siRNA/NTsiRNA and then treated with bleomycin/saline. The results were normalized by GAPDH (n = 3). (**E**,**F**) X-gal staining and quantification of SA-β-gal positive astrocytes (n = 5). The results were expressed as percentages of total cells. (**G**,**H**) ELISAs of IL-6 and IGFBP3 proteins in the culture medium (n = 4–6).

**Figure 5 cells-12-02008-f005:**
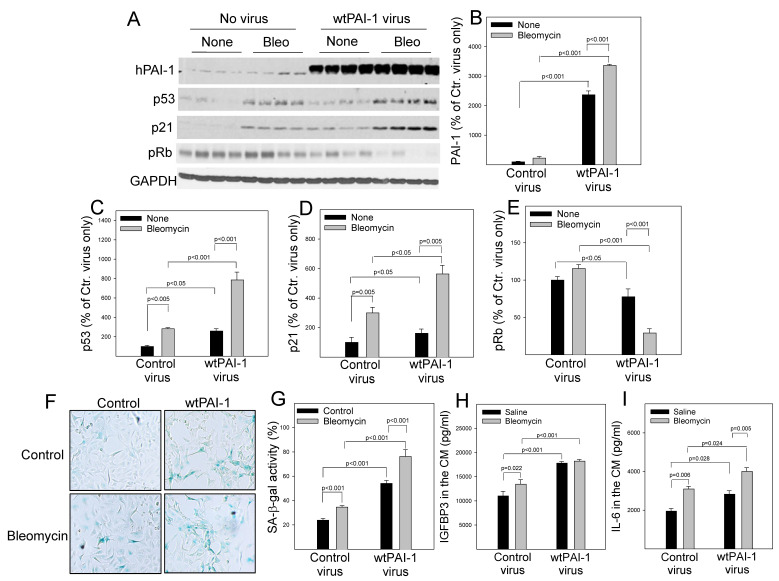
Overexpression of PAI-1 enhances bleomycin-induced ATII cell senescence. (**A**–**E**) Western analyses and quantification of PAI-1, p53, p21, and pRb proteins in PAI-1 and control lentivirus-transduced primary PAI-1^−/−^ ATII cells. The band intensities were normalized by GAPDH (n = 3–6). (**F**,**G**) X-gal staining and quantification of SA-β-gal positive cells; the results are expressed as percentages of total cells. (**H**,**I**) ELISA of IGFBP3 and IL-6 in the conditioned medium (CM, n = 6).

**Figure 6 cells-12-02008-f006:**
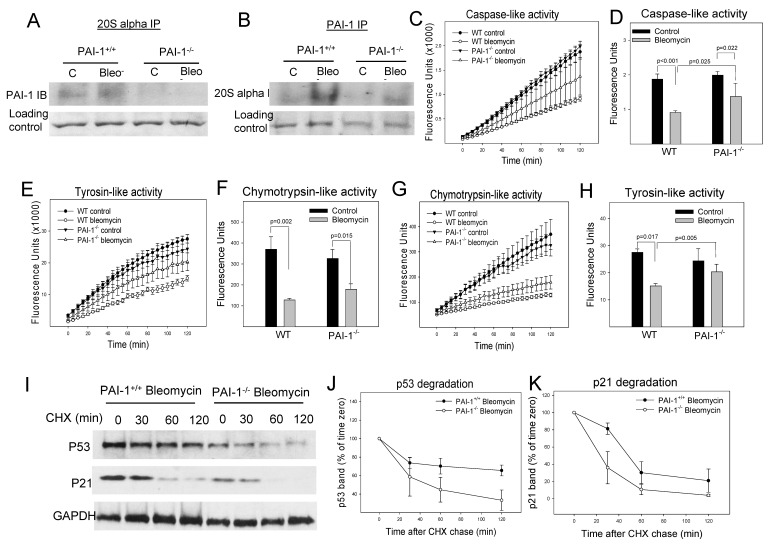
PAI-1 binds to proteasome components and inhibits proteasome activity and p53 degradation in primary mouse ATII cells. (**A**,**B**) Immunoprecipitation-immunoblotting analysis of PAI-1 interaction with proteasome 20Sα3 subunit. Primary ATII cells were isolated from wild-type (WT) and PAI-1 deficient (PAI-1^−/−^) mice and treated with bleomycin. PAI-1 and 20Sα3 proteins were immunoprecipitated with anti-mouse PAI-1 and anti 20Sα3 monoclonal antibody, respectively. Westerns were conducted with specific antibody to 20Sα3 or PAI-1. Ponceau S staining was performed to show equal sample loading. (**C**–**H**) Effects of overexpression of PAI-1 on the activities of proteasome enzymes. Primary ATII cells isolated from WT and PAI-1^−/−^ mice were treated with bleomycin, and the activities of three proteasome enzymes were assessed using special substrates as described in the method section. (**C**,**E**,**G**) Representative graphic pictures of fluorescence changes with time. (**D**,**F**,**H**) Quantitative data of three proteasome enzymes (n = 4–6). (**I**–**K**) PAI-1 suppresses p53 degradation in ATII cells. Primary ATII cells isolated from WT and PAI-1^−/−^ mice were treated with bleomycin for 24 h and cultured in bleomycin-free medium for another 48 h before being treated with cycloheximide, as described in the Materials and Methods section. The amounts of p53, p21, and GAPDH (loading control) were assessed via Westerns. The results were normalized by GAPDH and expressed as percentages of time zero (n = 3–4).

**Figure 7 cells-12-02008-f007:**
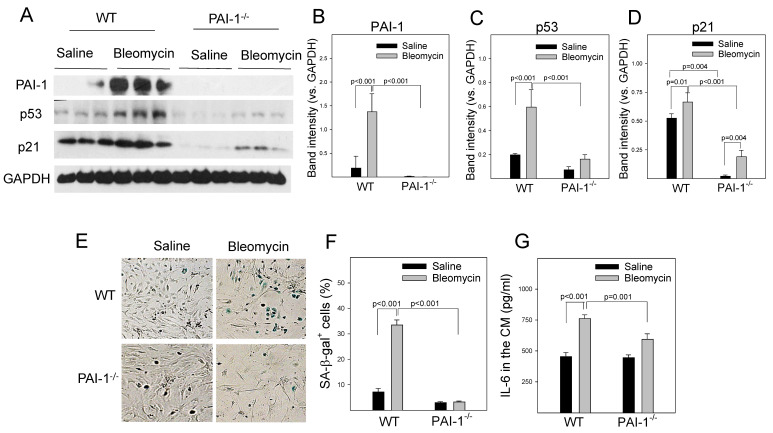
PAI-1 mediates bleomycin-induced ATII cell senescence. (**A**-**D**) Western analysis of the proteins of interest in ATII cells after bleomycin treatment. The band intensities of PAI-1. P53 and p21 were normalized by the corresponding GAPDH band (n = 3). (**E**,**F**) X-gal staining and quantification of SA-β-gal positive cells. The results were expressed as percentages of total cells (n = 6). (**G**) ELISA of IL-6 in the conditioned medium (CM, n = 6).

**Figure 8 cells-12-02008-f008:**
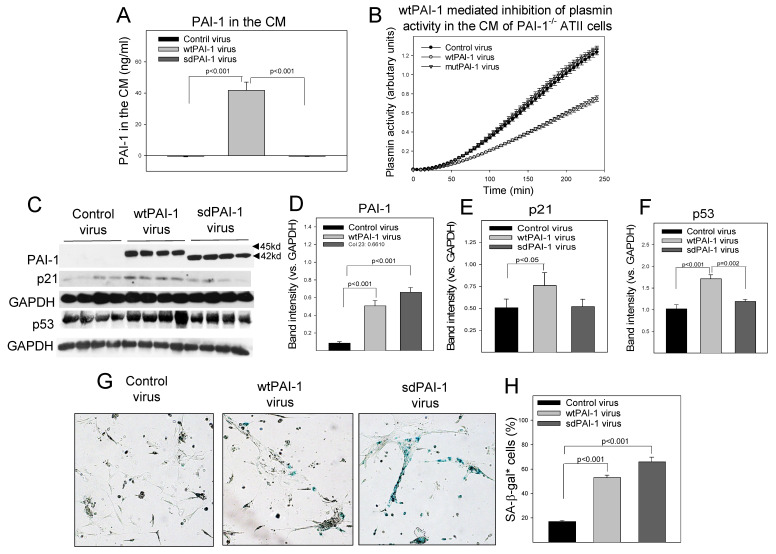
Effects of wild-type and secretion-deficient PAI-1 on p53 expression and senescence in primary ATII cells. (**A**) ELISA of PAI-1 protein in conditional medium (CM). (**B**) Plasmin activity in the CM was measured using a special fluorescence substrate. (**C**–**F**) Western analyses and quantification of PAI-1, p53, and p21 proteins in wtPAI-1, sdPAI-1, or control lentivirus-transduced PAI-1^−/−^ ATII cells. The band intensities were normalized by GAPDH (n = 4). (**G**,**H**) X-gal staining and quantification of SA-β-gal positive cells; the results are expressed as percentages of total cells (n = 6).

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
