# Peer review of "PAI-1 Regulation of p53 Expression and Senescence in Type II Alveolar Epithelial Cells"

_cells, 2023, doi:10.3390/cells12152008_

Round 1
Reviewer 1 Report
The current study by Tapsi Rana and colleagues investigates the role of plasminogen activator inhibitor (PAI-1) in regulation of type II alveolar epithelial (ATII) cell expression of p53 and senescence using bleomycin induced alveolar epithelial cell injury model. ATII cell renewal plays a major role in protection against fibrosis, and age-associated loss of ATII cell renewal capacity due to senescence predisposes to organ fibrosis, including idiopathic pulmonary fibrosis (IPF). Further, recent work from this group and others show that increased PAI-1 contributes to ATII cell senescence and apoptosis in correlation with induction of p53 and p53 transcriptional target p21. However, the underlying molecular mechanism by which PAI-1 contributes to increased p53 or senescence in fibrotic lungs, including IPF lung is not fully elucidated. The present study demonstrates that cellular and excreted PAI-1 inhibits proteosome activity by binding to the components of proteosomes in human lung adenocarcinoma (A549) cells and primary murine AII cells. This in turn leads to posttranslational stabilization of p53. The authors have used complementary immunoblotting, immunocytochemical and proteosome/enzymatic assays to draw the conclusion. Further, the authorship has compared the effects of forced overexpression of wild-type and secretion-deficient PAI-1 protein in wild-type and PAI-1 deficient ATII cells and siRNA mediated inhibition of bleomycin-induced PAI-1 expression to establish PAI-1 role in p53 induction and senescence. The current study is timely and establishes an interesting link between an increased PAI-1 expression and inhibition of proteasomal degradation leading to stabilization of p53 and senescence ATII cells. The study is interesting, novel and addresses an important question. The manuscript is well written and easy to understand. However, few minor changes are suggested to improve the overall contribution and presentation of the work.
Comments:
1. The rationale to use lung adenocarcinoma carcinoma (A549) cells for senescence related study is not clear.
2. Inhibition of PAI-1 using TM5275 in A549 cells exposed to bleomycin to understand the PAI-1 role in alveolar epithelial cell p53 expression and senescence during injury is reasonable. However, treatment A549 cells exposed lentivirus expressing wild-type PAI-1 with bleomycin is not clear and complicates direct demonstration of the PAI-1 role in p53 stabilization or senescence.
3. Results of immunoblotting clearly show that bleomycin injury further increased the binding of PAI-1 to 20Sa3 in PAI-1 overexpressing A549 cells (Figure 1D, lane 3 vs 4 upper panel). However, it is stated in lines 174-176, that “overexpression of wild-type PAI-1 protein alone increased the binding of PAI-1 to 20Sa3 and 19S Rpt3/S6b, although it did not further increase bleomycin-stimulated binding”. The authors need to revise their statement to be consistent with their results.
4. Published literature suggests that p53 mutation or loss of its expression rather than stabilization and activation often contributes to loss of p53 function in cancer cells. Therefore, statement in introduction (lines 43-45) “Although it is well documented that p53 undergoes a variety of posttranscriptional modifications that leads to stabilization and activation of this protein in cancer cells, how p53 is activated in ATII cells in fibrotic lung is unclear” needs revision. Same sentence is also repeated in the discussion (lines 396-398).
5. Some of the GAPDH blots are overexposed.
6. Reference 19, Yang et al and reference 37, Yahata are incomplete.
No major concern.
Reviewer 2 Report
The authors show evidence of the participation of PAI-1 in the stabilization of the p53 protein during the development of senescence in lung cells through the binding of PAI-1 to proteasome subunits and a decrease in its ability to destroy p53. The paper is of undoubted interest, in view of the discovery of a previously unrevealed mechanism of interaction between PAI-1 and the proteasome, however, there are a number of fundamental remarks.
The researchers claim to be working with ATII of the mouse lung. However, no characterization of this cell line is given for its authentication. The above protocol for the isolation of cells from the mouse lung makes it possible to obtain with the same success cell lines of lung fibroblasts, which proliferate much more actively and are able to displace ATII from the cell culture already after 1–2 passages. The paper does not indicate at what passages the cell studies were performed, and the microscopic images in Figures 5, 7, and 8 are of very poor quality, but allow one to discern the fibroblast-like morphology of the cell line, which the authors of the work unreasonably call ATII. Thus, the entire central hypothesis of the work, centered around the aging of lung stem cells, is called into question if the authors do not provide convincing data to authenticate this lineage. As well there are no information presented about A549 cell line origin.
The materials and methods section is written carelessly, the manufacturers of most reagents, the instruments and software used, the origin of plasmids and viral constructs, the composition (or manufacturer) of the lysis buffer for the Western are not indicated, the quantities are for some reason spelled out in letters: Fifty micrograms [96], Two hundred micrograms [120], Ten micrograms [134]. It is not described how the conditioned medium for the ELISA assay was prepared. It is not specified which second antibodies were used for the western.
The quality of the illustrations presented is not satisfactory. The resolution of the fonts used does not allow you to see what is indicated in small print. Authors must provide images of adequate quality that will allow the reader to assess the reliability of the data presented. Figures 4, 5, 7 and 9 contain images of b-gal of insufficient resolution, which do not contain a description of the microscopic method, the magnification used, etc.
The presentation of the work must be greatly improved before it can be published.
